# Analysis of Trace Volatile Compounds Emitted from Flat Ground and Formed Bed Anaerobic Soil Disinfestation in Strawberry Field Trials on California's Central Coast

Kali Prescott [1], Stefanie Kortman [1,†], Josue Duque [1,2,†], Joji Muramoto [3,4], Carol Shennan [4], Gloria Greenstein [1] and Arlene L. M. Haffa [1,*]

1  Department of Biology and Chemistry, California State University, Monterey Bay, Seaside, CA 93955, USA; kprescott@csumb.edu (K.P.); skortman@csumb.edu (S.K.)
2  Department of Life & Environmental Sciences, University of California, Merced, Merced, CA 95343, USA
3  Center for Agroecology, University of California, Santa Cruz, Santa Cruz, CA 95064, USA
4  Environmental Studies Department, University of California, Santa Cruz, Santa Cruz, CA 95064, USA
*  Correspondence: ahaffa@csumb.edu; Tel.: +1-831-582-4695
†  These authors contributed equally to this work.

**Abstract:** Anaerobic soil disinfestation (ASD) is emerging globally as an alternative to fumigant pesticides. To investigate ASD mechanisms, we monitored microbially produced volatile organic compounds (VOCs) and other volatile gases in situ using Fourier Transform Infrared Spectroscopy. Study plots infested with *Fusarium oxysporum*, *Macrophomina phaseolina*, and/or *Verticillium dahliae* included: organic flat ground (fASD, 6.7 + 13.5 megagrams per hectare, Mg/ha, rice bran/broccoli) and uncovered soil treated with mustard seed meal (MSM, 3.4 Mg/ha) at one site performed in fall of 2018; formed bed (bASD, 20 Mg/harice bran), control (UTC) and fumigant (FUM) at a second field site in fall of 2019 and 2021. Here, we present VOC diversity and temporal distribution. fASD generated 39 VOCs and GHGS, including known pathogen suppressors: dimethyl sulfide, dimethyl disulfide, and n-butylamine. bASD produced 17 VOCs and greenhouses gases (GHGs), 12 of which were also detected in fASD but in greater concentrations. Plant mortality and wilt score (fASD: 3.75% ± 4.79%, 2.8 ± 0.8; MSM: 6.25% ± 12.50%, 2.7 ± 0.3; bASD: 61.27% ± 11.26%, 4.1 ± 0.5; FUM: 13.89% ± 7.17%, 2.3 ± 0.2; UTC: 76.63% ± 25.11%, 4.3 ± 1.0) were significantly lower for fASD and MSM versus bASD and UTC ($p < 0.05$). Only FUM was not statistically different from fASD and MSM, and was significantly lower than UTC and bASD (bASD-FUM, $p < 0.05$; UTC-FUM, $p < 0.05$). The cumulative strawberry yield from bASD-treated plots was not different from FUM or UTC (bASD: 60.3 ± 13.6; FUM: 79.4 ± 9.19; UTC: 42.9 ± 12.4 Mg/ha). FUM yield was significantly greater than UTC ($p = 0.005$). These results, and to a far greater extent, additional challenges faced during both bASD trials, suggest that bASD is not as effective or as feasible at maintaining overall plant health as fASD or traditional fumigants. However, differences in management practices and environmental conditions at both sites across years cannot be fully excluded from consideration and many of our observations remain qualitative in nature.

**Keywords:** strawberry; alternatives to fumigant pesticides; Anaerobic Soil Disinfestation (ASD); volatile organic compounds; *Fusarium oxysporum* f. sp. *fragariae*; *Verticillium dahliae*

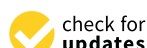



## 1. Introduction

California supports 90% of the nation's $2.8 billion strawberry industry [1]. In Monterey County alone, more than 4000 hectares are in production. The industry has traditionally relied on chemical fumigants to limit loss due to pathogens [2]; however, in 2016, California completed the total phase-out of methyl bromide to protect the ozone layer per the Montreal Protocol. It is one of the most toxic and widely used soil fumigants. Previously, California's strawberry production used 18% of the methyl bromide in the United States.

This phase-out has significantly shifted industry pest management techniques [3]. Other fumigant pesticides have become popular; however, additional public pressure is pushing the industry to reduce or eliminate their use, leaving crops vulnerable to rapidly spreading destructive soilborne pathogens. There is an urgent need to develop other effective pathogen suppression methods that reduce the risk to public health and the environment.

Anaerobic soil disinfestation (ASD) is proposed as an alternative management strategy to some traditional fumigants [4–7]. Pest management via ASD works by establishing an anaerobic environment that shifts the soil microbial populations to obligate and facultative anaerobes [8–10]. The practice begins with the addition of an organic labile carbon amendment to facilitate increased microbial activity during the community shift. Next, gas exchange and oxygen supply are limited via the application of an impermeable plastic mulch, and the soil pores are saturated with water.

Once anaerobic, microbial fermentation produces greenhouse gases (GHGs) such as carbon dioxide ($CO_2$), methane ($CH_4$), nitrous oxide ($N_2O$), volatile organic compounds (VOCs), and other volatiles which can inhibit fungal plant pathogens such as *Verticillium dahliae*, *Macrophomina phaseolina*, and possibly also *Fusarium oxysporum*. All three pathogens are of major concern to strawberry growers on the Central Coast of California and can be easily introduced [6,11–13]. *Verticillium dahliae* and *F. oxysporum* are both characterized by wilted foliage and stunted plants while *M. phasolina* causes crown rot. All three pathogens result in the die-off of older leaves and can lead to crown collapse and the eventual death of the entire plant [14–16]. Traditionally, fumigants, such as methyl bromide combined with resistant cultivars, have been the primary management method to address pathogen presence in these systems.

The microbially produced VOCs (mVOCs) believed to be key players in ASD are small organic compounds (<300 Da in mass or <15 carbons in length) with high vapor pressure and a low boiling point. They are also generally lipophilic and readily diffuse into air or water-filled pores in the soil depending on polarity [17–19]. Significant evidence from studies of ASD suggests that mVOCs are the primary cause of pathogen suppression [11,12,20–25]. Furthermore, the study of soils inhospitable to certain pathogens reveals more diverse mVOC profiles associated with an increased abundance of biocontrol microorganisms [10,26,27]. This suggests that mVOCs may serve as indicators of biocontrol organisms or directly inhibit the pathogen growth cycle. However, more research including the possible effects of GHG production is needed to understand the whole pathogenic system under ASD treatment.

Previous studies have been limited in their ability to study the complete profile of mVOCs produced over the course of ASD and are thus unable to characterize profile differences between different methods of ASD. An extensive profiling of mVOCs may help to delineate the mechanisms that are occurring and inform ASD users on which method may be more appropriate for their system. A more comprehensive understanding of the mVOCs produced in response to changes in abiotic characteristics such as soil moisture, pH, and temperature, and by biotic factors such as the application of organic matter, may be necessary [17,19,28,29].

In coastal California strawberry production, flat ASD (fASD) has traditionally been used in the summer, while formed bed ASD (bASD) has been the main approach for fall [6]. fASD consists of the ASD treatment applied to a leveled soil surface where the carbon source has been folded into the soil and the treatment area is covered with Vapor Safe Totally Impermeable Film (TIF™, Raven Industries, Sioux Falls, SD, USA). bASD involves incorporating the carbon source into the soil as the soil is mounded and shaped into the long, continuous beds in which the crops will eventually be planted. Once the beds are formed, either TIF or plastic mulch is wrapped over each bed, leaving the soil in the furrows between beds exposed. If TIF is applied, this covering will be removed and replaced with opaque plastic mulch before the crop is planted. Growers may prefer bASD because of its potentially lower cost than fASD; namely, due to the associated labor and equipment needed to prepare the field for strawberry planting after fASD, and taking

land out of production to implement the several-weeks-long summer fASD treatment. Fall bASD allows for the grower to plant a cash crop in the summer and then treat the field in the fall when it would normally be fumigated. Additionally, because many growers own a bed mulching attachment, bASD can be more readily assimilated and is more easily translated from traditional fumigation in terms of pre-crop preparation. However, cumulative elevated temperature is a critical mechanism required for the success of ASD, and this may be harder to achieve in the fall. To our knowledge, VOC production in summer bASD has not been characterized, and this study may present a middle-ground alternative for growers to consider.

To increase the understanding of molecular mechanisms behind ASD effectiveness, we designed a method to monitor VOCs and other gases in situ from both fASD and bASD used for pest control in strawberry crops on the California Central Coast. Here, we present the VOCs and other gases that we identified that were produced during both types of ASD and provide a method for comprehensive VOC detection. We present data for mortality, wilt, and strawberry yield for each treatment to provide context for the efficacy of each ASD method against specific plant pathogens and the VOCs that were present during the treatment period.

## 2. Materials and Methods

### 2.1. Field Trial Sites and Experimental Design

This study was conducted across two field sites over the course of three years: one trial at an organic farm at the University of California Santa Cruz, Center for Agroecology (UCSC-CfA) in August 2018, and two trials at a commercial farm Plant Sciences Incorporated (PSI) in Watsonville, CA in August of both 2019 and 2021. During all trials, under-tarp emissions were measured every 1–2 weeks from ASD plots and control plots (when available) and at PSI in 2021, traditional fumigation plots were also measured. Soil temperature and moisture probes (5TE, Meter Group, USA) and oxidation-reduction potential (ORP) sensors (Sensorex Solutions, Garden Grove, CA, USA) were installed to a depth of 15 cm and connected to data loggers (CR1000, Campbell Scientific, Logan, UT, USA) for continuous data acquisition. Soil reduction potential (Eh) values from ORP sensors were converted to standard hydrogen electrode output by adding 199 mV to each result. When results dropped below +200 mV, the soil was considered to be anaerobic. The duration the Eh values remained below +200 mV was used as a predictor for ASD efficacy, as well as the soil temperature at 20 cm, as all of our field sites were infested with *F. oxysporum*. We were unable to replicate our treatments across all sites and years. As such, the efficacy of treatments is determined based on within-site differences between treatments.

### 2.2. Flat ASD Field Trial

The first trial was conducted from 2018 to 2019 at an organically managed farm at the University of California Santa Cruz, (UCSC-CfA, 36.982504, −122.056449). The region is characterized by a Mediterranean climate with warm, dry summers and cool, wet winters. The mean annual temperature ranges from 7 °C in the winter to 21 °C in the summer, with an annual average precipitation of 31 cm. The soil is Elkhorn sandy loam (fine-loamy, mixed, thermic pachic argixerolls) and is naturally infested with *V. dahliae*, *F. oxysporum*, and *M. phaseolina* [30]. The experiment was a split-split plot design with four replicates, and is described in detail by Michuda et al. (2019) and Zavatta et al. (2021) [30,31]. Each plot was 3.7 m wide and 6.9 m long consisting of four 0.93 m wide (center-to-center) beds. The middle 2 beds of each plot were used for data collection during the strawberry crop period. We monitored from "control" uncovered plots treated with 3.4 Mg/ha mustard seed meal (MSM), and from fASD plots with a carbon source of 15 Mg/ha rice bran and 5.2 Mg/ha crop residue. The site had undergone a four-year broccoli rotation prior to the trial.

Carbon sources were applied by broadcasting and then incorporated into 15.2 cm soil depth with a spader along with any remaining crop residue. The plots were briefly leveled and compacted with a ring roller. Drip tapes (~0.64 cm per hour) were laid with 15.2 cm

spacing. The fASD plots were covered with clear totally impermeable film (TIF) plastic mulch (VaporSafe, Viaflex, Sioux Falls, SD, USA), and irrigated by drip tape within 48 h of the carbon source incorporation (Figure 1a). MSM plots were left uncovered (Figure 1b).

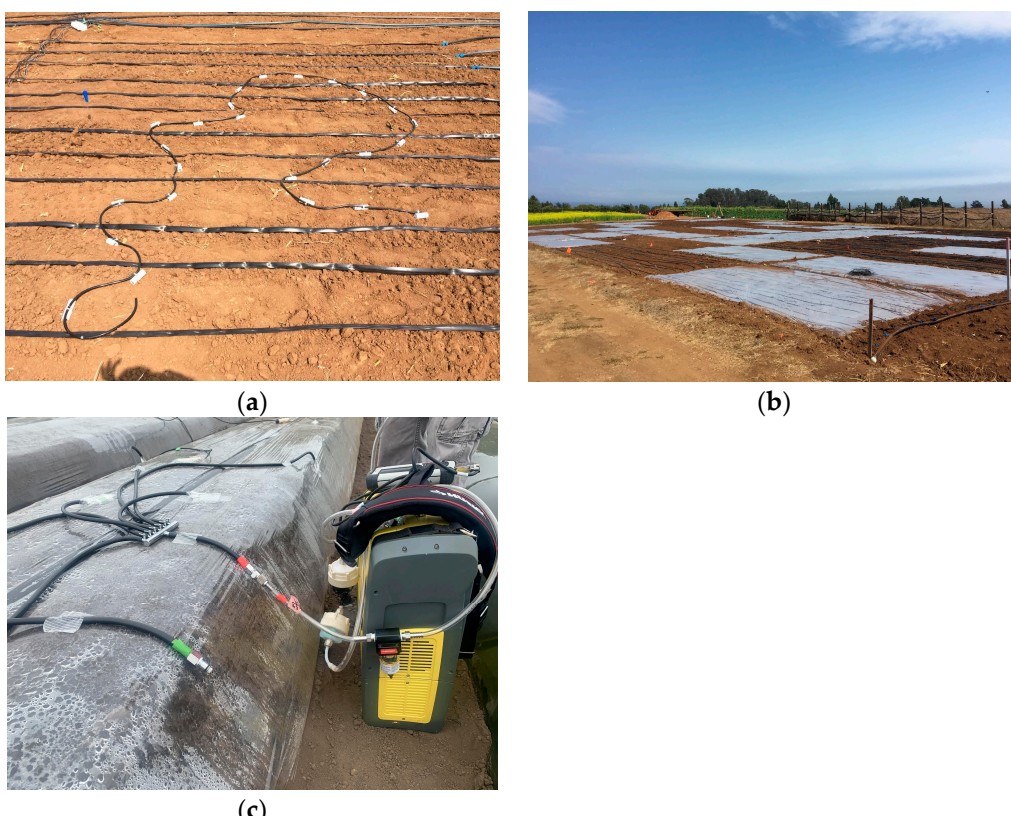

(a)

(b)

(c)

**Figure 1.** Types of ASD treatments established for this study. (**a**) Perforated tubing laid over the plots and drip tape during fASD at UCSC-CfA. The tubing was laid out to allow headspace gas to be sampled across the central interior of the plot rather than from a single localized point. This system also allowed for a broad sampling of the headspace near the center of the plot without having to disturb the soil. (**b**) fASD at UCSC-CfA with the clear TIF plastic laid over the treatment plots along with the various open-air control and treatment plots. (**c**) bASD at PSI with the tubing manifold connected to a DX4040 FTIR Gas analyzer collecting gas spectra. Hoses of varying lengths were laid over the tarp and inserted into the tarp at random points along the bed to allow for sampling of the headspace gas across a larger area of the bed using a manifold.

To measure VOCs for fASD plots in situ, we used portable Fourier-transform infrared (FTIR) Gas Analyzers (DX4040, Gasmet Technologies, Vantaa, Finland) described in detail below. We monitored gases under the tarp from one location at each of the four replicate treatment plots. Before the tarp was laid out, opaque, UV-resistant soft (Durometer 70A) tubing (6 m length, 6.4 mm dia.) was perforated every 30 cm and placed on top of the soil and positioned to cover as much of the plot area as possible. This would act as the inlet line for sampling with the DX4040 FTIR Gas Analyzer. A shorter length of tubing (0.5 m) was placed on top of the soil with the end pointing away from the general area of the perforated tubing to act at the outlet or return line for the DX4040. Once the tarp was in place and sealed, the ends of both lengths of tubing were pulled through slits made in the tarp. We sealed the slits with clear Gorilla tape that covered the holes and wrapped around the tubing to create a seal. We inserted a barbed, stainless steel quick-disconnect coupling with automatic shut-off at each end of the two lengths of tubing that were on the outside of the tarp. During monitoring events, the quick-disconnect ends of the tubing were connected with the corresponding inlet and outlet lines of the DX4040. Given that the MSM plots were not tarped, a modified static chamber method was used to analyze

gas headspace samples [32,33]. The chambers were modified to have quick connections similar to the covered ASD beds so that gases could be sampled by the FTIR analyzers directly from the chambers. Gases were sampled from three randomly selected locations within each plot. While fASD was sampled weekly, MSM was sampled biweekly on a separate day.

The plastic TIF tarp remained in place for 6.4 weeks and soil moisture was kept above field capacity for three weeks. The TIF tarp was removed on day 44, and the beds were shaped and covered in opaque mulch. Planting holes were cut on day 75 and Albion cultivar strawberries were planted on day 81. Disease incidence was monitored by weekly tracking of plant mortality in each plot and visually scored assessment of wilt. The crop was harvested weekly to biweekly from April to September and the marketable and non-marketable yields were recorded [30]. The cumulative yield was determined from the summation of the marketable yield from each harvest and the final plant mortality at the end of the harvest.

### 2.3. Formed Bed ASD Field Trials

Trials two and three were conducted from 2019 to 2020 and 2021 to 2022, on a conventional field site in partnership with Plant Sciences Incorporated (PSI), Watsonville, CA. The region is characterized by a Mediterranean climate with warm, dry summers and cool, wet winters. The mean annual temperature ranges from 8° to 20 °C with an average annual precipitation of 61 cm. The soil type is classified as Elder sandy loam. PSI suffered from Fusarium wilt of strawberries in the 2017–2018 season.

The field was organized in a randomized complete block design (RCBD) with four blocks and three treatments: bASD, tri-form 80 (79.8% Chloropicrin and 19.5% 1,3-Dichloropropene) fumigant (FUM), and an untreated control (UTC) replicated within each block for a total of 12 sampling locations. The bASD carbon source (20 Mg/ha rice bran) was broadcasted on flat ground and incorporated to 15.2 cm with a rototiller before the beds were listed and shaped. High-flow irrigation tape (Sumitomo 3 GPM/100 feet) was laid at the center of each bed top. All beds were covered with clear TIF film. Irrigation began within 48 h of the carbon source incorporation. Each treatment in each block was applied to two adjacent beds (1.22 m wide × 0.37 m tall × 15.2 m long). For PSI 2019, one bed was designated for a randomized block trial in which the bed was subdivided into quadrants with each quadrant assigned randomly to one of four strawberry varieties (Albion, San Andreas, Royal Royce, or Valiant). The second bed was divided into two sections with the back field section planted with Plant Sciences Inc. variety PS1160 and the front field section planted with PS9271; this second bed was not monitored for volatile gas emissions, wilt, mortality, or yield. The RCBD was repeated for PSI 2021 for the pre-planting treatments, but not the randomized strawberry varieties.

For monitoring gases under the tarp in formed beds, a modified design of the under-tarp tubing from UCSC-CfA was installed. Rather than a single long length of perforated tubing, six lines of varying lengths (two each of 60, 90, and 120 cm) of opaque, UV-resistant tubing were connected to a six-port manifold and extended along beds to various locations across the top and sides of the beds, covering a total area of 1.7 m$^2$. Tubing was secured in place above the soil surface, but not touching the soil, with Gorilla clear tape applied where the tube extended through the hole in the TIF. A 15 cm length of tubing was connected on one end to the single manifold outlet port, and a quick-disconnect coupling was connected to the other end for integrating with the DX4040 inlet line. A 0.5 m length tubing was placed under the tarp for attaching to the outlet line of the DX4040 as described above to ensure a non-destructive sampling by redepositing sampled gases back into the system (Figure 1c). The tubing was only installed in the beds that were designated for randomized strawberry variety planting.

For the PSI 2019 trial, we measured gases under the TIF tarp in the summer and early fall for bASD-treated plots only. Sampling did not occur from days 6 to 18 due to a field closure after fumigation. We did not initially monitor control plots with FTIR for this trial

because, unlike MSM at UCSC-CfA, the PSI control plots were tarped with TIF. However, during the PSI 2021 trial, the tarped control plots (UTC) were monitored in tandem with bASD and FUM to compare across treatments. The PSI 2021 bASD trial never achieved anaerobicity and sampling was discontinued after day 43. Thus, PSI 2021 bASD data are not reported, only results from the FUM and UTC treatments which continued to be monitored for the remainder of the treatment periods.

The bASD treatment duration was 12.1 weeks for both PSI trials. Soil moisture was kept above field capacity for 3 weeks. When the TIF was removed from the PSI 2019 trial, the beds were covered in opaque film within 48 h. Holes were punched a week later and the strawberry crop was planted a week after that. In 2021, the TIF was not replaced and holes were punched at the end of the bASD treatment. Multiple strawberry varieties were planted at PSI in both years, but the crop yield data for the Albion cultivar grown in 2019 are reported because it was planted at both UCSC-CfA and PSI 2019. Due to the failure of bASD in 2021 and labor shortages, the grower opted to plant *F. oxysporum*-resistant cultivars in both the bASD and UTC (San Andreas) and a less resistant cultivar in FUM plots (Monterey). For the 2019 PSI trial, disease incidence was tracked through weekly monitoring of plant mortality in each plot and weekly visual assessment of wilt score. Cumulative marketable fruit yield was estimated in megagrams per hectare. Plant mortality and yield were tracked during the 2021 trial but not the wilt score.

### 2.4. Gas Collection and Analysis

Portable Fourier-transform infrared (FTIR) Gas Analyzers (Gasmet DX4040, Gasmet Technologies, Vantaa, Finland) were used to analyze VOCs. The FTIR gas analyzers were calibrated on-site at the beginning of each sampling day with compressed nitrogen gas (99.999% $N_2$ purity, Linde, Watsonville, CA, USA). For all tarped plots and treatments, spectrum samples were collected and a baseline analysis was performed by the FTIR analyzers using a non-destructive replacement method. First, the sample-in hose of the FTIR was connected to the sample line of the tarp or inlet port on the modified static chamber for un-tarped plots. The sample-out hose and sample-out ports were left disconnected, allowing the FTIR to pump air through the detection cell and expel the residue gases in the chamber into the atmosphere. Once the air inside the Gasmet analytical cell achieved equilibrium with the air under the tarp or chamber and spectra were stable (approximately 4 min), the second hose that recirculated collected gases into the sampling apparatus was connected. Only measurements taken after the sample-out hose was connected were recorded. The data were logged using the Calcmet Lite software (Gasmet Technologies, Vantaa, Finland) and loaded onto a personal data assistant (PDA) to receive data from the DX4040 analyzer via a hardwired connection or Bluetooth.

The FTIR Gas Analyzer is sensitive to water vapor and condensation on the internal gold mirrors must be strictly avoided. It was recommended to keep the proportion of water vapor below 3% to protect the mirrors; however, we found that reliable identification of VOCs requires keeping maximum water vapor at 1% or lower. To do this in the extremely moist environment under the tarp, a water trap was constructed and placed on the inlet line to the FTIR. The trap was made of a series of two filter housings filled with 3A molecular sieve clay bead desiccant (Delta Adsorbents, Roselle, IL, USA) and a 50 μm filter (Figure 1c). The desiccant was changed out between plots or as needed and the FTIRs were flushed with compressed $N_2$ gas when necessary to remove excess moisture. The filter paper was changed when discolored or excessively moist.

Spectra were collected every 2 min for 12–40 min at each treatment plot one to two times a week during ASD. Control plots were less frequently monitored due to resource constraints.

### 2.5. FTIR Spectra Analysis and Data Limitations

Sample spectra were identified and analyzed in the Calcmet Pro Software version 12.18 and compared to a pre-built library of 375 reference spectra compiled by Gasmet. Libraries were then developed for each unique sample set to determine the approximate concentra-

tion of each gas in the sample. Sample spectra were also compared to the NIST/EPA vapor phase IR library in the Calcmet database to identify potential novel gases. Those novel gases were added to the reference spectra if sufficient references were available.

Uncertainty of the quality and resolution of available reference spectra for many VOCs necessitated a minimum limit of accurate detection of 1 ppmv (per recommendations from Gasmet Technologies). Any concentrations that measured greater than 0 but less than 1 ppmv were excluded; however, some VOCs were only detected at concentrations below 1 ppmv. Given that the concentration of the VOC is not necessarily correlated with its effectiveness at pathogen suppression, these VOCs are reported qualitatively to promote future research.

Lastly, the libraries provided by Gasmet and the NIST library are incomplete. Any gases identified by Calcmet Pro that returned a fit lower than 90% were disregarded from the search criteria entirely. However, it is possible that some compounds were not detected when present (false negative) or misidentified (false positive) by Calcmet due to similarities between the spectrum and the low resolution of certain reference spectra. For example, many organic acids cannot be reliably detected using FTIR in high moisture environments as their spectra are easily obscured by water, and many alkanes may be difficult to distinguish from one another, particularly in anaerobic environments with higher concentrations of methane.

### 2.6. Data Analysis of Gas Concentrations

For gases that occurred with regular frequency in both treatments at UCSC-CfA specifically, the MSM plot values were averaged and standardized for the dates where sampling of the control plots did not occur. This allowed for pair-wise comparison between standardized values in the MSM and the fASD-treated plots for each sample day. This was necessitated by reduced sample frequency at the MSM-treated plots due to time constraints created by the static chamber sampling method. MSM and fASD could not always be sampled on the same days. This was not the case for the UTC or FUM plots at either PSI trial as those plots were tarped and sampled at a similar frequency to the bASD. Repeated measures of two-way ANOVA followed by Tukey post-hoc analyses were performed on the gas concentrations measured from the UCSC-CfA and PSI field sites to detect differences over time and between treatments using R studio version 4.1.0 (RStudio Team, 2022).

A secondary analysis using two-way repeated measures ANOVA was performed to compare the gas concentrations between just the bASD and fASD treatments and between the Eh phase (aerobic and anaerobic) within each treatment. As much of the VOC data contain missing values due to non-detection or measurements which did not meet the 1ppmv accurate detection tolerance, a Linear Mixed Effects model was employed as a non-parametric alternative where necessary. In all models, the replicate plot identifier was included as a random effect. We feel a measure of confidence in the within-site comparisons between treatments; however, the lack of replication across sites and over years greatly reduces our confidence in the comparisons between treatments that occurred at different sites and in different years. Furthermore, many of the VOCs that we measured did not result in a great enough sample size above the 1ppmv detection threshold to perform a quantitative analysis. As such, those results and any differences between sites are presented qualitatively. The code for this analysis is available in a GitHub Repository.

### 3. Results

#### 3.1. Monitoring of Anaerobic Conditions during ASD

Eh and soil temperature values showed that the UCSC-CfA fASD achieved and maintained anaerobic conditions and high internal soil temperatures necessary for successful fASD (Figure 2a). All of the fASD plots fell below +200 mV within the first 24 h of tarping. Anaerobic conditions were maintained for 23, 30, 38, and 17 days for each of the four plots, respectively. Soil temperature at 20 cm was 29.7 °C $\pm$ 2.9 °C (Figure 3a), which was

significantly greater than the MSM (22.8 °C ± 1.4 °C; two-sided *t*-test, *p*-value < 0.05), and 29.0 °C ± 1.9 °C at 30 cm.

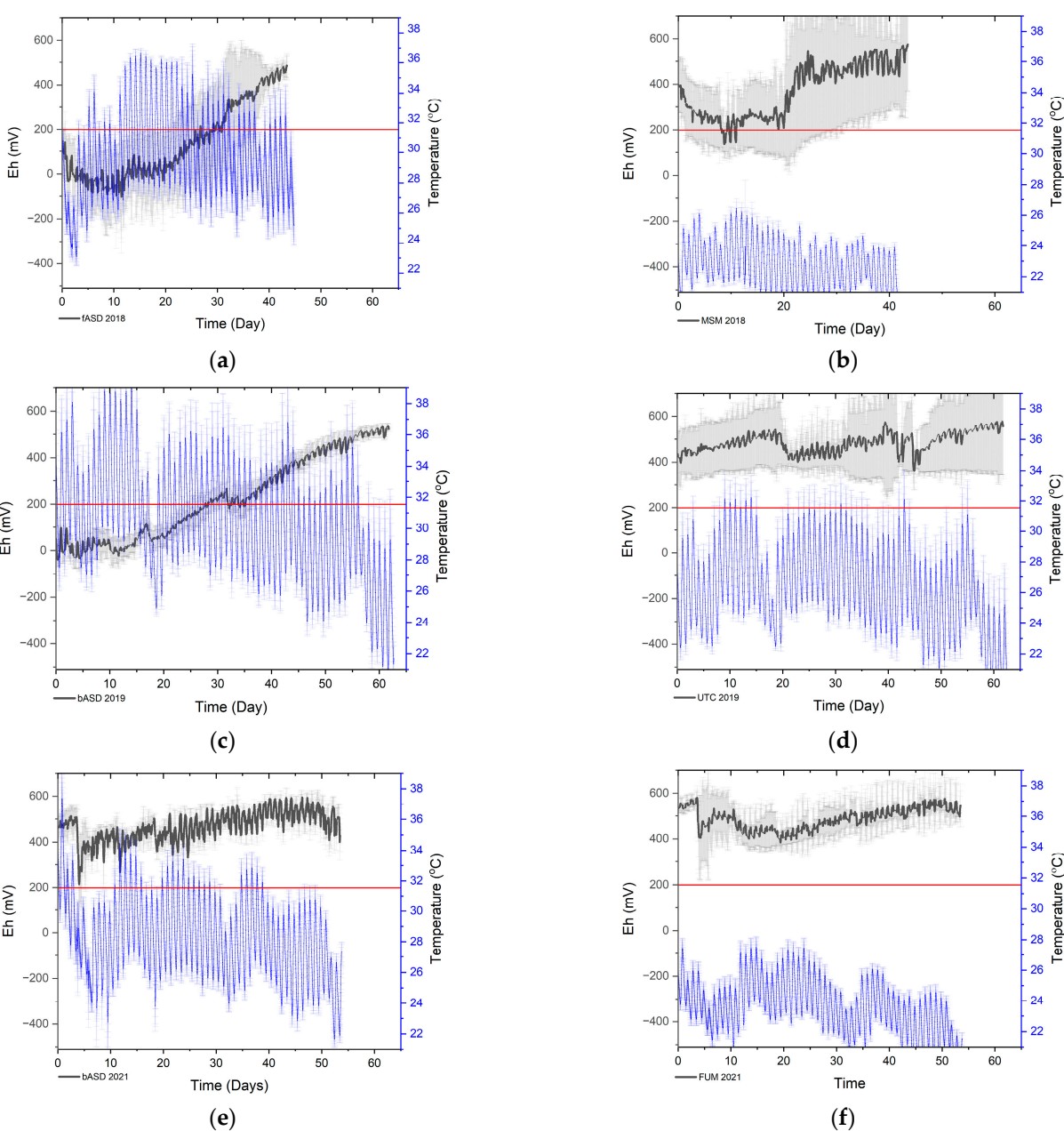

**Figure 2.** The redox potential (Eh) and temperature of all of the ASD-treated replicates show that results varied somewhat between replicates but significantly between trials. The red line indicates the target Eh threshold for anaerobic soil conditions. (**a**) The fASD trial at the UCSC-CfA was aerobic by day 30 across all plots and was significantly warmer than (**b**) the MSM. The bASD PSI 2019 trial (**c**) became also aerobic after 30 days and was also significantly warmer than the UTC (**d**). However, the PSI 2021 bASD trial (**e**) was only anaerobic for a short period (~24 h); there was no difference in Eh compared to the FUM in 2021 (**f**). All comparisons between the ASD treatments across all three trials were significantly different in temperature with PSI2019 being the warmest but most variable at 30.5 °C ± 3.8 °C, and PSI 2021 being the coolest at 28.2 °C ± 2.7 °C.

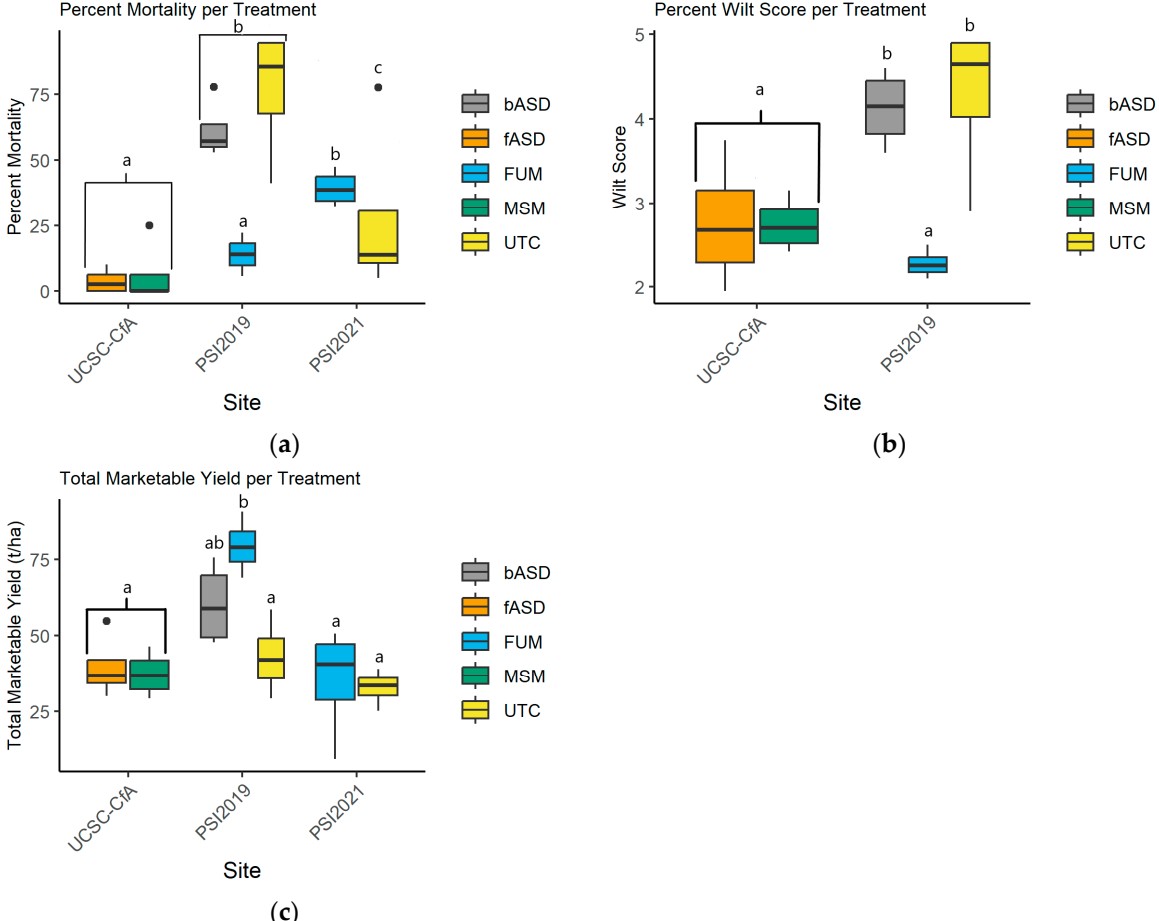

**Figure 3. (a)** Both treatments at UCSC-CfA (fASD and MSM) had significantly lower mortality than all other treatments with the exception of the fumigated treatment in the 2019 trial; although, we cannot separate the effect of treatment from site. The mortality of UTC was likely most influenced by the strawberry variety with Fusarium wilt-susceptible Albion in 2019 seeing high mortality and Fusarium wilt-resistant San Andreas seeing much lower mortality in 2021. **(b)** Wilt score followed a similar trend to mortality with reduced wilt overall at UCSC-CfA and in the FUM treatment, but not in the bASD and UTC treatments. Wilt score was not collected for PSI 2021 because the grower changed the strawberry variety from Albion to San Andreas. **(c)** Cumulative marketable yield of strawberries harvested from the crops grown after ASD was measured for each treatment at both sites and compared with the cumulative yield of strawberries in t/ha$^{-1}$ at the UCSC-CfA. There was only a marginal difference in yield between the fASD and MSM at UCSC-CfA. At PSI 2019, bASD was noticeably less successful with reduced yield in all ASD plots compared to FUM. FUM had a significantly higher yield than the UTC with bASD having no significant difference between either the FUM or UTC. The yield results for the PSI 2021 are reported but there was no difference between treatments as Fusarium wilt-resistant varieties were planted in the non-FUM treatments. Significant differences between treatments are indicated by lower case letters with same letters indicating no differences.

The 2019 bASD trial at PSI maintained anaerobic conditions and achieved internal bed temperatures of 30.5 °C ± 3.8 °C at 20 cm which was significantly higher than the UTC beds (26.3 °C ± 3.0 °C; two-sample *t*-test, *p*-value < 0.05) (Figure 3b). Eh values dropped below +200 mV within the first 72 h during the 2019 formed bed trial. Anaerobic conditions were maintained for 22, 24, 29, and 30 days for each of the four plots, respectively. The duration of the 2019 ASD treatment at PSI was 41 days longer than at UCSC-CfA. On day 71 it was determined that the beds were above the +200mV threshold long enough to be considered aerobic (Figure 2b). Prior to day 71, it was noted that the beds were drying out beneath the TIF leading to the beds partially collapsing under the tarp. Additionally,

tractor activity in the field caused large tears in the tarp which exposed the soil to air for an unknown period and had to be repaired.

For the 2021 trial, by day 33, Eh values had failed to drop below the +200 mV threshold for a sufficient enough duration to establish anaerobic conditions. This bASD application was determined to have failed as the beds were anaerobic for barely 24 h (Figure 2c). The bed temperature at 20 cm was on average 28.2 °C ± 2.7 °C (Figure 3c) and 27.2 °C ± 1.6 °C at 30 cm and was significantly higher than the fumigant (23.6 °C ± 1.7 °C; two-sample *t*-test, *p*-value < 0.05). The bASD soil temperature at 30 cm was significantly lower than the bASD soil temperature at 20 cm (paired *t*-test, *p*-value < 0.05).

Comparison of the soil temperature at 20 cm across trials revealed that the fASD trial was significantly cooler than the 2019 bASD trial, but both the fASD and 2019 bASD trials were significantly warmer than the 2021 bASD trial (One-Way ANOVA, *p*-value < 0.05).

### 3.2. Evaluating ASD Efficacy on Plant Health and Yield

For the strawberry crop following ASD, at UCSC-CfA, the fASD treatment had lower plant mortality (3.75 ± 4.79) and wilt score (2.8 ± 0.8) compared to MSM (6.25 ± 12.5 and 2.7 ± 0.3, respectively) (Figure 3a,b). However, this was not statistically significant per separate Two-Way ANOVAs with interactions for wilt score and mortality between treatment and between sites. At the PSI 2019 trial, plant mortality and wilt score were lowest in FUM (13.9 ± 7.2 and 2.3 ± 0.2) compared to bASD (61.3 ± 11.3 and 4.1 ± 0.5) and UTC (76.6 ± 25.1 and 4.3 ± 1.0); the latter, having the highest mortality and wilt score. Both plant mortality and wilt score were significantly greater during bASD and UTC compared to the fASD and MSM treatments (*p* < 0.05). However, the plant mortality and wilt score of the FUM treatment at PSI 2019 were not significantly different from either fASD or the uncovered MSM treatment. The mortality of the 2021 fumigated treatment was not different from any of the treatments in the previous trials or the 2021 UTC (Figure 3a), but the 2021 UTC mortality was significantly lower than the UTC in the 2019 trial (*p*-value = 0.04). The wilt score was not monitored for the PSI 2021 trial.

The mean cumulative marketable yield was compared with a Two-Way ANOVA with interactions between treatments and between sites. At UCSC-CfA, there was no significant difference between the fASD (39.6 Mg/ha ± 10.7) and the uncovered MSM (37.3 Mg/ha ± 7.6) (Figure 3c). Yields for fASD and MSM were not significantly different from PSI 2019 bASD (79.4 Mg/ha ± 9.2) or UTC (42.9 Mg/ha ± 12.4), but were significantly lower than the PSI 2019 FUM treatment (FUM-fASD, *p*-value = 0.006; MSM-FUM, *p*-value = 0.004). The bASD was not significantly different from either FUM or UTC, but FUM was significantly greater than UTC (UTC-FUM, *p*-value = 0.02). The ANOVA was performed with and without the PSI 2021 data for FUM and UTC and there was no change in the significance of the across-site comparisons. Although yield and mortality data were collected for the PSI 2021 FUM and UTC treatments, the grower's decision to change from Albion to Monterey in the FUM treatment and Albion to San Andreas in the UTC means that any differences in these data cannot be separated from the differences between varieties. Furthermore, due to the lack of replication across sites, the impact of the site on wilt score, mortality, and yield cannot be separated from the impact of treatment alone.

### 3.3. Gas Analysis

#### 3.3.1. UCSC-CfA fASD Trial

Thirty-nine gases (including VOCs and GHGs) were identified at UCSC-CfA. Nineteen gases were identified in the MSM plots, thirteen of which were also detected in the fASD: carbon dioxide, methane, nitrous oxide, butylamine, heptane, pentane, propene, hexylamine, 1-pentene, propylene oxide, heptane, 1-hexanol, hexene. The remaining gases were only detected in MSM: nonane, undecane, hexane, cyclohexane, isopentane, and 1,3,5-trimethylbenzene (Figure 4; mean measured concentrations summarized in Table 1). Of the compounds detected in both treatments, only 1-pentene and propylene oxide showed a

significant difference in concentration between the fASD treatment and MSM (Repeated measures ANOVA *p*-value < 0.05).

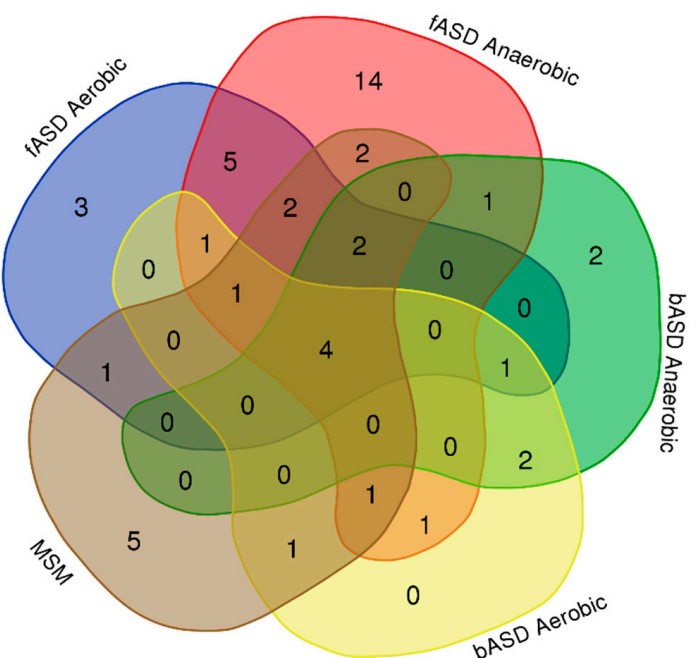

**Figure 4.** Venn diagram presenting the total number of volatile gases detected during field trials that included ASD monitoring (UCSC-CfA and PSI 2019) and according to the anaerobic and aerobic phases. UCSC-CfA overall produced the greatest quantities and greatest variety of gases between the two trials. Meanwhile, only four gases ($CO_2$, $N_2O$, $CH_4$, propylene oxide) were shared across all phases of fASD, bASD, and the MSM. fASD overwhelmingly produced more gases and the majority of those were produced during the anaerobic phase with some detected in the aerobic phase as they declined. Only four compounds were detected that were unique to bASD: 1-butanethiol, 2-methylpyrazine, cis-2-butene, and cis-2-pentene. Undecane was the only gas unique to both the aerobic phase of bASD and MSM. Five compounds were unique to the MSM: nonane, cyclohexane, 1,3,5-trimethylbenzene, hexane, and isopentane. UTC and FUM were not included in this diagram as they are both from the PSI 2021 treatment when bASD failed before the end of the trial period. The gas concentrations for UTC and FUM are summarized in Table 1. Figures were generated with an open-access web tool at https://bioinformatics.psb.ugent.be/webtools/Venn. URL (accessed on 5 February 2023).

Twenty-six additional compounds were identified in the fASD-treated plots which were not detected in the MSM plots. The compounds detected in the fASD were further divided according to the soil Eh during the sampling event. Each replicate plot was anaerobic by day 2 of the treatment period. However, each plot became aerobic at different periods starting at day 17 with all plots reaching aerobic Eh values by day 38. Fourteen gases were detected during the anaerobic phase of the fASD, while three were detected only during the aerobic phase, and fifteen were detected during both phases (Figure 4). Carbon dioxide, nitrous oxide, methane, propylamine, 2-ethylhexanol, hexanoic acid, isohexane, tetrahydrothiopene, isobutanol, t-butanol, 1-pentanol, methyl chloride, 2,3-heptanedione, ethyl fluoride, 1-methylimidazol, nitrogen dioxide, ethanolamine, 1,2,3-trimethylbenzene, 2,3-hexanedione, dimethyl sulfide, and dimethyl disulfide were either only present in the anaerobic phase or significantly greater during the anaerobic phase than the aerobic phase (Repeated-measures ANOVA, *p*-value < 0.05 or Linear mixed model with repeated measures, *p* < 0.05).

**Table 1.** Mean values of gases detected in each treatment. fASD during the anaerobic phases has generally greater mean concentrations for nearly all gases with some exceptions. The mean value of these gases was generally lower in the aerobic phase for both sites and lower at PSI overall. For means less than 1 ppm, if the gas was detected in the treatment, then the concentration fell to 0, and those data were treated as true 0′s when calculating the mean; however, any values between 0 and 1 ppm were excluded.

| Gas Species | fASD Aerobic | fASD Anaerobic | MSM (2018) | bASD Aerobic (2019) | bASD Anaerobic (2019) | FUM (2021) | UTC (2021) |
|---|---|---|---|---|---|---|---|
| 1,2,3-Trimethylbenzene | | 0.842 | | | | 1.475 | 0.666 |
| 1,3,5-Trimethylbenzene | | | 1.320 | | | | |
| 1-Butanethiol | | | | 5.203 | 6.042 | | |
| 1-Butene | 1.730 | 5.706 | 1.158 | 1.102 | | 0.654 | |
| 1-Heptene | 1.055 | 1.080 | | 0.319 | | | |
| 1-Hexanol | 1.439 | 2.303 | 1.174 | | | 0.103 | |
| 1-Methylimidazol | | 7.672 | | | | | |
| 1-Pentanol | | 1.966 | | | | | |
| 1-Pentene | 1.475 | 1.306 | 0.538 | | 0.938 | 0.175 | 0.178 |
| 2,3-Dimethylpyrazine | 1.290 | | | 0.571 | 0.029 | | |
| 2,3-Heptanedione | | 0.417 | | | 0.163 | | |
| 2,3-Hexanedione | | 1.221 | | | | | |
| 2-Ethylhexanol | | 1.895 | | | | | |
| 2-Methylpyrazine | | | | 0.237 | 0.238 | 0.073 | |
| 3-Chloro-2-methyl-1-propene | 1.190 | | | | | | |
| 4-Ethyltoluene | | | | | | | 1.505 |
| 4-Methyl-3-penten-2-one | 1.154 | | | | | | |
| Allylcyanide-3-butenenitrile | 1.312 | | | | | | |
| Ammonia | | | | | | 0.023 | 0.229 |
| Butylamine-1-butanamine | | 5.842 | 1.799 | 2.845 | | | |
| Carbon dioxide | 49,690 | 82,123 | 883 | 10,335 | 48,227 | 12,863 | 11,952 |
| Carbon monoxide | 0.129 | 0.733 | 0.001 | | | | |
| Cis-2-Butene | | | | | 0.205 | | |
| Cis-2-Pentene | | | | | 1.020 | | |
| Cyclohexane | | | 2.830 | | | | |
| Dimethyl disulfide | 0.875 | 1.468 | | | | | |
| Dimethyl sulfide | 5.868 | 3.952 | | | | | |
| Dodecane | 7.082 | 1.090 | | | | | |
| Ethanol | 4.2 | 41.0 | | | | | |
| Ethanolamine | | 0.793 | | | | | |
| Ethyl chloride | | | | | | 0.861 | 0.855 |
| Ethyl fluoride | | 15.957 | | | | | |
| Heptane | | 12.927 | 2.652 | | | | |
| Hexane | | | 0.428 | | | | |
| Hexanoic acid | | 5.646 | | | | | |
| Hexene | 4.566 | 0.998 | 0.838 | | 1.040 | 1.464 | 0.197 |
| Hexylamine | 2.930 | 1.260 | 1.250 | | | | |
| Isobutanol | | 1.530 | | | | | |
| Isohexane | | 36.378 | | | | | |
| Isopentane | | | 1.247 | | | | |
| Methane | 1545.48 | 679.86 | 1.82 | 1.78 | 15.87 | 1.68 | 1.71 |
| Methyl chloride | | 20.347 | | | | | |
| M-Xylene | | | | | | 0.639 | |
| Nitrogen dioxide | | 13.308 | | | | | |
| Nitrous oxide | 9.54 | 8.34 | 0.34 | 33.82 | 5.38 | 4.09 | 2.07 |
| Nonane | | | 5.937 | | | | |
| O-Cresol | | | | | | 0.603 | 0.584 |
| Pentane | | 1.113 | 2.538 | | | | |
| Propene | 1.399 | | 1.153 | | | | |
| Propylamine | | 0.797 | | 0.080 | | | |
| Propylene oxide | 3.192 | 4.148 | 0.444 | 0.036 | 0.963 | 2.038 | 0.374 |
| t-Butanol | | 2.050 | | | | | |
| Tetrahydrofuran | | | | | | | 0.233 |
| Tetrahydrothiopene | 1.910 | | | | | | |
| Undecane | | | 1.374 | 9.532 | | | |

In both fASD and MSM, greater concentrations of gases and other compounds occurred in the first two and a half weeks of the experiment before declining. This was also largely during the anaerobic phase. The number of different compounds detected on a given



sample day increased after the first 2 weeks of the trial and decreased over the third and fourth weeks of the trial. Measurements on days 19 and 26 showed those to be the most active days for volatile gas production as the greatest number of individual compounds were detected on those days. This coincided with fASD plots transitioning one by one to the aerobic phase and may be explained by the soil community activity changing in response to oxygen availability.

The highest concentrations for eighteen gases were measured on day 2 at the very onset of the anaerobic phase. $CO_2$, $N_2O$, and $CH_4$, and propylene oxide were the most consistently measured compounds during fASD followed by 1-pentene, 1-hexanol, and 2,3-hexanedione. A significant spike in ethanol production (326.78 ppmv) occurred in the first 24 h after the TIF was laid and the plots were irrigated, then ethanol concentrations decreased below the limit of detection over the next 3 days. Overall, gas production declined during the aerobic phase except for dodecane which increased significantly.

### 3.3.2. 2019 PSI bASD Trial

The volatile gas profile of the bASD plots in the 2019 PSI trial was unique compared to the UCSC-CfA trial, where seventeen gases (including VOCs and GHGs) were detected in bASD. Similar to fASD, ethanol was detected in the first 24 h of sampling once the plots were sealed with the TIF; however, the peak was less than 1 ppmv. Anaerobicity was achieved in the first two days.

Similar to the fASD trial, all four replicated plots in the bASD became aerobic by day 30. Unlike the fASD, the bASD plots became aerobic within the same 24 h. Also different from fASD is the distinct grouping of gases during bASD according to the anaerobic and aerobic phases, and the lack of observed temporal trends in gas concentrations and detection that were present in fASD.

Five gases were detected only during the anaerobic phase of bASD, and five during only the aerobic phase; seven gases were detected during both phases. Cis-2-butene, cis-2-pentene, hexene, 1-pentene, and 2,3-heptanedione were only present during the anaerobic phase of bASD, while propylamine, butylamine, 1-butene, undecane, and 1-heptene during the aerobic phase (Figure 4). Of those gases present during both phases, 2-methylpyrazine and 1-butanethiol were significantly greater during the anaerobic phase (Linear Mixed Model: *p*-value > 0.05), and 2,3-dimethylpyrazine was not different between phases. Propylene oxide was detected in both phases with no significant difference. GHGs were measured in both soil phases with $CO_2$ and $N_2O$, being greater in the aerobic phase; however, there was no difference in $CH_4$ between phases. Both propylene oxide and the GHGs were present in concentrations lower than at UCSC-CfA; $N_2O$ and $CO_2$ were significantly lower (Repeated Measures ANOVA; F-stat = 4.859, *p*-value = 0.0278; F-stat = 16.78, *p*-value = 0.0064).

## 4. Discussion

### 4.1. Challenges of FTIR

A novel technique was used to measure VOCs under ASD conditions using a portable FTIR Gas Analyzer. While employing this technique, we encountered many advantages and disadvantages which we have detailed. It is important to stress that FTIR gas analysis at this current time is not a replacement for gas chromatography, which remains the gold standard for gas analysis. FTIR has been used to measure GHGs with accuracy comparable to gas chromatography due to the availability of high-quality reference spectra. As of current, the quality of reference spectra available for many compounds is less reliable; thus, necessitating the need for a 1 ppmv detection limit.

The DX4040 can accurately measure GHGs and VOCs greater than 1 ppmv when water vapor is less than 3% volume but ideally less than 1% volume as water vapor can obscure the spectra of some gases. Organic acids are difficult to detect if there is any water vapor in the sample at all. Preventing water vapor from exceeding these quantities can be extremely challenging to achieve when monitoring an extremely wet environment such

as under the tarp during ASD. Through vigilant use of desiccant-filled water traps and microfilter paper, our team was able to maintain water vapor below the 1–3% volume necessary for detecting GHGs and VOCs; however, we were not able to dry the gas samples enough to accurately monitor the production of organic acids. Thus, we are unable to assess the speculation by Momma (2008) and Momma et al. (2013) that organic acids may have contributed to pathogen suppression [10,34]. It is also extremely important to note that in addition to water vapor, the quality of the sample spectrum is easily compromised by high concentrations of $CO_2$ and $CH_4$ that occurred in quantities great enough to obscure lower-concentration VOCs with similar spectra.

Considering these limitations, we were still successful in obtaining detailed VOC profiles at both field trials spanning the entire duration of the ASD treatment. VOCs were cross-referenced in the VOC database and with other primary literature, and VOCs linked to pathogen suppression and biocontrol organisms, such as *Clostridium* sp. and *Enterobacter* sp., were identified with a notable degree of confidence by the researchers. These findings, relative to ASD efficacy, were in line with additional data collected by our collaborators including yield, soil temperature, and redox values. Thus, we are confident that some general conclusions can be drawn from VOC measurements using FTIR. However, more specific questions may require more specialized methods and measuring techniques such as gas chromatography.

This paper serves to provide information about the functional groups and VOCs produced during ASD using the most robust spectra available; however, additional analysis is required to confirm the presence of these compounds at lower concentrations (<1 ppmv). Reference libraries are not yet detailed enough to measure VOC concentrations with the same degree of certainty as gas chromatography; however, FTIR is superior in its ability to measure a broad suite of gases without the need for destructive sampling.

### 4.2. Gas Identification and Evidence of Biocontrol Organisms

A vast array of volatile organic compounds and greenhouse gases was detected under the tarp during anaerobic soil disinfestation. Most of the compounds detected were simple hydrocarbons of a variety of functional groups, and were detected at higher concentrations earlier in the trials when the system was anaerobic during both ASD treatments and the fumigant. The higher concentrations were also likely in response to bioavailable C amendments. Concentrations declined as the trials progressed, as available C and N were exhausted. By day 30 at both sites, the treatments became aerobic, and the majority of the VOCs had or were beginning to drop below detectable concentrations. The GHGs ($CO_2$, $CH_4$, and $N_2O$) also declined as the ASD treatment became aerobic but stayed well above ambient until the tarps were pulled. A number of other organic compounds, sulfides, and oxides were also detected. However, many of the VOCs detected were below the 1 ppmv detection limit recommended by the manufacturer of the DX4040 FTIR Gas Analyzers. The detection limit is established due to the limited availability of high-resolution spectrum for VOCs. We include all of the identified compounds that returned a match probability of 90 percent or more, but it should be noted that compounds with concentrations below 1 ppmv have a higher margin of error and a higher probability of being misidentified by the Calcmet software.

The gas profiles at the UCSC-CfA and PSI trials were distinctly unique; however, many gases were shared across both trials, though not necessarily by the corresponding treatments at each trial. Carbon dioxide, methane, nitrous oxide, and propylene oxide were the most consistently detected gases and were the only compounds detected in all treatments at both trials. $CO_2$, $CH_4$, and $N_2O$ are common greenhouse gases with high-quality references, so measurement data is more reliable. Propylene oxide was likely produced by the TIF, drip tape, and other plastic equipment used during the trial that may have degraded upon exposure to UV light.

We cross-referenced all of the VOCs identified in the trials with the VOC database [35]. We found numerous VOCs that are associated with biocontrol organisms such as *Clostridium* sp.

and *Enterobacter*, or themselves act as biocontrol agents. These include: DMS, DMDS, ethanol, isobutanol (*Alternaria alternata (Fr.), Keissler* in co-culture with *F. oxysporum*), 1-propanol, and 1-pentanol (*Verticillium longisporum*) [26,36–40]. 2,3-dimethylpyrazine was detected during both fASD and bASD over multiple days, and has been linked to several bacteria genera including: *Serratia* sp. and *Stenotrophomonas maltophilia* [41,42]. However, 2,3-dimethylpyrazine has also been linked to the plant pathogen *Pseudomonas aeruginosa* [38].

In both the fASD and bASD, butylamine-1-butanamine was detected; which, is produced by Proteus mirabilis, a widely abundant facultative bacteria found in soil [43]. High concentrations of butylamine-1-butanamine later in the trials is consistent with a microbial shift to anaerobic metabolism; although, this compound was also present in the MSM treatment at UCSC-CfA, but in lesser quantities. Several VOCs were identified in both ASD treatments and are known metabolites of *Clostridium difficile*, a proposed biocontrol organism [44–49], including: 1-butanethiol, dimethyl disulfide, dimethyl sulfide, butanol, 1-pentanol, 1-hexanol, and methylcyclopentane (<1 ppmv). Elevated concentrations of these volatiles may indicate increased abundance or increased metabolic activity of *Clostridium* sp. Several other VOCs detected were not in the database. Further studies on the microbial community may help to determine if any of the remaining VOCs that we detected have microbial origins.

### 4.3. Efficacy of fASD and bASD

To assess the efficacy of fASD and bASD, we evaluated the treatments by their VOC and GHG profiles including the presence of specific gases associated with pathogen suppression. We evaluated ASD success by monitoring soil temperature and soil anaerobicity during both ASD trials and plant health throughout the crop and compared them to the within-site control plots (fASD:MSM or bASD:UTC:FUM). We also consider challenges in management between the two techniques and the feasibility of application within commercial fields. Overall, we found that fASD continues to be a more reliable method for ASD; however, bASD may be feasible given further refinement.

The 2019 bASD trial had an overall greater yield, which was expected as conventional strawberry systems usually have higher yields than organic [50]; however, at UCSC-CfA, we saw better overall plant health. Mortality and wilt score were similar between the fASD and MSM plots, but both treatments at UCSC-CfA were significantly lower than the bASD and UTC treatments at PSI 2019. Only the FUM treatment at PSI 2019 was comparable to UCSC-CfA in regard to overall plant health and survival. This would suggest that management practices or other factors are more likely the reasons for the improved plant health at UCSC-CfA rather than fASD specifically. Although, previous fASD treatments investigated at UCSC-CfA generally showed higher yield and less evidence of wilt compared to fallow or mustard seed cake-only applications [30]. However, our monitoring of VOCs, soil temperature, and soil redox potential suggests that ASD was less effective at PSI and likely a result of the formed bed method, which may be due in part to low internal bed temperatures inhibiting disinfestation.

The differences in VOC profiles that we observed between ASD treatments suggested that there may be less microbial activity in the soils of bASD compared with fASD. The first indicator of reduced microbial activity was our observation that bASD produces fewer VOCs and lower concentrations of both VOCs and GHGs as compared to fASD. The comparatively lower concentrations of both $CO_2$ and $N_2O$ suggest reduced microbial activity during bASD. Only fifteen VOCs were identified from the bASD performed at PSI in 2019 compared to thirty-seven VOCs detected in one trial of fASD performed at UCSC-CfA. Over twenty additional compounds were detected at PSI and six at UCSC-CfA which had to be excluded from analysis as they never breached the 1 ppm limit of accurate detection. This suggests that more VOCs may have been present at both sites, but overall concentrations were still lower in bASD.

When specifically investigating known pathogen suppressors, DMS and DMDS, we found there to be less presence of these compounds in bASD versus fASD, which may

indicate differential success for the two ASD methods. In bASD, DMS was only detected once and DMDS was never detected during any monitoring events. In contrast, DMS and DMDS were repeatedly detected throughout the fASD treatment at UCSC-CfA during the anaerobic phase (Figure 5). Although these treatments are difficult to compare directly, the reduced plant health of the bASD and UTC treatments at PSI compared to the FUM strongly suggests that ASD treatment was less effective. Ethanol, another volatile known to indicate ASD success, was also observed to vary notably between the ASD treatments [23,51]. Ethanol was detected in the greatest quantities (326.78 ppmv) during the first 24 h of the fASD trial and in smaller quantities during the 2019 bASD trial (>1 ppmv). No ethanol was detected in the failed bASD trial at PSI in 2021. Lower production in bASD of gases associated with pathogen suppression, DMS and DMDS, and the ASD success indicator ethanol, could suggest that soil microbial activity in the bASD trials was reduced as compared to our fASD trial. However, this could also be due to better overall soil health at UCSC-CfA. To determine why flat ground may be a more feasible method for grower use of ASD, we also considered the challenges in the implementation of these practices.

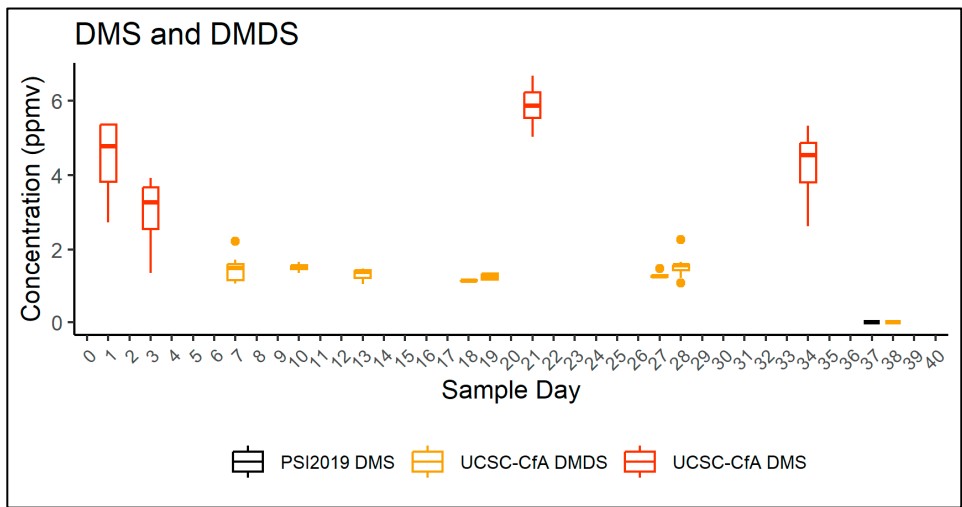

**Figure 5.** DMS and DMDS were detected repeatedly during the fASD trial. DMS and DMDS were never detected from the mustard seed cake control, so were not reported. DMDS was notably absent from the PSI 2019 trial. DMS was only detected once at very low concentrations during the entire ASD treatment period.

A closer look at the feasibility of each ASD method required that we investigate the capacity for growers to maintain anaerobic conditions and soil temperature above 30 °C when deploying fASD or bASD. The variation in oxidative reductive potential across replicates in the fASD is attributed to animal activity in the field (gophers and coyotes) which led to more air penetration into the plots. Although we observed that soil redox potential seemed more consistent for bASD, with all beds becoming anaerobic by the end of day 1 and remaining anaerobic until day 30 ± 24 h, we found that maintaining certain conditions for ASD on formed beds was difficult. In our first bASD trial in 2019, bed temperatures were significantly warmer than fASD, but were more variable. The increased variability may have been due to differences in local climate during the trial; however, bed integrity was continuously problematic during the 2019 bASD trial. Beds tended to dry out more easily, and sealing the TIF at the base of the beds was difficult. Some beds began to collapse inside the TIF, causing unpreventable bulging and tears. This lack of integrity made maintaining soil saturation challenging and may have led to more variability in soil temperature. The bASD in 2021 was deemed a failure when the soil redox potential became aerobic on day 4 ± 48 h and remained aerobic for the duration of the trial. Additionally, the soil temperature at 20 cm was significantly cooler than the 2019 bASD trial and the fASD trial.

Previous studies indicate that to control *Fusarium oxysporum* f. sp. *fragariae* by fASD, soil temperature at 20 cm depth must be above 30 °C for 450 h or more [52]. However, the results of this study suggest that bASD may require 30 °C soil temperature throughout the depth of the bed to be successful. We did not install soil temperature sensors at depths greater than 20 cm during the 2019 bASD trial. As the beds were 16.6 m by 1.3 m with a depth of 0.36 m, the temperature of the soil in the bed core was not monitored. The internal temperature of the beds may not have reached temperatures high enough to facilitate disinfestation, which is further indicated by the lower gas production at this trial. Temperature sensors were installed at 20 cm and 30 cm during the 2021 trial. The soil temperature was overall lower in 2021 than during all other trials, but notably, the temperature at 30 cm was significantly lower than the temperature at 20 cm (1 °C difference). Surviving pathogen communities in the bed core may have been able to recolonize closer to the surface of the bed post-ASD, leading to significantly greater mortality and wilt score in the ensuing strawberry crop.

Although we observed significant differences between the trials, unfortunately, the effects of soil temperature cannot be statistically separated from the effects of fASD and bASD as we were unable to replicate each method across sites. Thus, we cannot rule out that historic management practices at PSI may have contributed to the soil community response to bASD and the reduced gas production. However, the failure to maintain anaerobic conditions during the second attempted bASD trial in 2021 and our difficulty to manage the integrity of the beds throughout the 2019 trial would lead us to strongly consider the unreliability of the bASD method. Further refinement of bASD may solve a number of problems we encountered, but based on the present management technique, we would conclude that ASD in formed beds may not be a successful method for Fusarium pathogen suppression.

In our fASD trial, we did not experience the same challenges associated with bed integrity and soil temperature as bASD. Although we did encounter challenges in maintaining the integrity of the TIF in this trial due to wildlife activity, it was manageable within the small experimental field. The most obvious challenge we encountered with fASD was the spatial constraints. At PSI, the many strawberry varieties being grown in close proximity for the trial made it logistically impractical to perform fASD as the grower would have to sacrifice too much acreage to accommodate the fASD plots and navigate shaping beds for the fumigant treatments. This likely would not be a constraint in a large, uniformly managed commercial field. To reach sufficient temperatures to disinfest *F. oxysporum*, fASD must be performed in the summer; which, conflicts with summer vegetable crops, and also requires allowing for the saturated soil to dry out and then forming and tarping of the beds closer to when strawberries are planted. Thus, farmers are more inclined to use fall bASD, but run the risk of not achieving sufficiently high soil temperature to adequately disinfest the soil. This may present a significant logistical transition for some growers. However, bASD would allow farmers to shape and list beds many weeks in advance of planting, and studies have shown that this approach can be successful in the management of *V. dahliae* which dies off at lower temperatures [4]. bASD may align more closely to fumigation management practices where fields are prepped with listed and tarped beds for fumigation a few weeks before planting. We plan to reattempt ASD in a commercial field using what we have learned from this trial. In doing so, we hope to expand FTIR analysis for both ASD and fumigated treatments.

**Author Contributions:** Conceptualization, A.L.M.H., J.M., C.S. and S.K.; methodology, S.K., K.P. and J.D.; software, K.P. and J.D.; validation, J.M., S.K. and K.P.; formal analysis, K.P. and A.L.M.H.; investigation, S.K., K.P., J.D., G.G. and J.M.; resources, S.K., A.L.M.H., J.M. and C.S.; data curation, K.P., A.L.M.H., S.K. and J.M.; writing—original draft preparation, K.P.; writing—review and editing, K.P., S.K., G.G. and A.L.M.H.; visualization, K.P. and A.L.M.H.; supervision, A.L.M.H.; project administration, A.L.M.H.; funding acquisition, A.L.M.H., S.K., J.M., C.S. and K.P. All authors have read and agreed to the published version of the manuscript.

**Funding:** This research was funded by the California State University Agricultural Research Institute (CSU ARI) with matching funds from the United States Department of Agriculture (USDA) grant number 2017-51181-26832.

**Data Availability Statement:** Data available on request due to restrictions e.g., privacy or ethical. The data presented in this study are available on request from the corresponding author.

**Acknowledgments:** We would like to acknowledge Plant Science Inc. for being our industry partner and allowing the use of their land for both PSI trials in 2019 and 2021, and the University of California Santa Cruz Center for Agroecology and Sustainable Food Systems for their facilitation and participation in the 2018 field trial. Additionally, we would like to acknowledge the students who helped collect field data: Armando Flores, David Gonzalez, Liliana Lee, Nicole Lucha, Israel Mandujano Olivera, Lauren McMullen, Joshua Ploshay, Elise Vasquez, and Sydnee Winbush.

**Conflicts of Interest:** The authors declare no conflict of interest. The funders had no role in the design of the study; in the collection, analyses, or interpretation of data; in the writing of the manuscript; or in the decision to publish the results.

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
