# Peer review of "Analysis of Trace Volatile Compounds Emitted from Flat Ground and Formed Bed Anaerobic Soil Disinfestation in Strawberry Field Trials on California’s Central Coast"

_agronomy, doi:10.3390/agronomy13051190_

Round 1

Reviewer 1 Report

The study presented in this paper investigates the mechanisms of anaerobic soil disinfestation (ASD), which is becoming a popular alternative to fumigant pesticides. The authors monitored microbially-produced volatile organic compounds (VOCs) and other volatile gases in situ using Fourier Transform Infrared Spectroscopy. The study plots were infested with various pathogens and treated with different ASD methods. The authors found that fASD generated the most VOCs, including known pathogen suppressors, and had the lowest plant mortality and wilt score. The authors also found that the yield of bASD was not significantly different from FUM or UTC, but FUM had a significantly greater yield than UTC. Overall, this study provides valuable insights into the mechanisms of ASD and its potential as an alternative to fumigant pesticides.While, the manuscript is need some minor modifications:

1:Is there any data of 2022 ?

2:standardize the Boxplots.

Author Response

1:Is there any data of 2022 ?

The study was not repeated for 2022. 

2:standardize the Boxplots.

I am uncertain what the reviewer is referring to here. The boxplots represent 3 different sets of data: yield, mortality, and wilt score which have different units or are unitless. Boxplot colors are coordinated by treatment. Furthermore, wilt score was not collected for PSI 2021, the reasons for which are explained in the methods and throughout the paper. I have edited the boxplot to include letter notation for statistical significance and updated the caption to include more details.

Reviewer 2 Report

This work presents relevant information about the potential of anaerobic soil disinfestation (ASD) for the control of pathogens in strawberries. As an important methodological contribution, the identification, and quantification of VOCs are carried out using Fourier Transform Infrared Spectroscopy, and the advantages and disadvantages of the technique are discussed. 

My main observation regarding this work has to do with the experimental design and statistical analysis. Not all treatments were evaluated under homogeneous conditions, but rather they were divided into groups that were evaluated in different areas or times. The effect of treatments could be confused with the effect of soil or climate factors that vary between the different places where the different treatments were evaluated. I consider it essential that the authors analyze this possibility and include the pertinent considerations in this regard in the text.

Lines 17,30,56: Write scientific names in italics

Line 31: I think this table is not necessary as the authors explain the meaning of each abbreviation the first time they use it.

Line 346: Since the treatments UTC and fASD together with MSM were evaluated in two separate areas (UCSC-CfA and PSI), ¿how the authors are sure that when comparing the treatments located in UCSC-CfA with those located in PSI, there are no confounding effects, which can be attributed to differences between the two experimental areas and not to the treatments?

Line 347: Did you mean Figure 3 a and 3 b?

Line 352: Again,  how the authors are sure that when comparing the treatments located in UCSC-CfA (MSM) with those located in PSI (fASD and FUM), there are no confounding effects, which can be attributed to differences between the two experimental areas and not to the treatments?

Line 360: How the authors are sure that when comparing the treatments located in UCSC-CfA with those located in PSI 2019, there are no confounding effects, which can be attributed to differences between the two experimental areas or time periods and not to the treatments?

Some additional comments are included in the attached file.

Author Response

This work presents relevant information about the potential of anaerobic soil disinfestation (ASD) for the control of pathogens in strawberries. As an important methodological contribution, the identification, and quantification of VOCs are carried out using Fourier Transform Infrared Spectroscopy, and the advantages and disadvantages of the technique are discussed. 

My main observation regarding this work has to do with the experimental design and statistical analysis. Not all treatments were evaluated under homogeneous conditions, but rather they were divided into groups that were evaluated in different areas or times. The effect of treatments could be confused with the effect of soil or climate factors that vary between the different places where the different treatments were evaluated. I consider it essential that the authors analyze this possibility and include the pertinent considerations in this regard in the text.

Lines 17,30,56: Write scientific names in italics

I have gone through the paper and attempted to correct this were ever it was lacking as well as correct some inconsistencies when referring to Verticillium dahliae the species and Verticillium wilt the disease.

Line 31: I think this table is not necessary as the authors explain the meaning of each abbreviation the first time they use it.

I have removed the table.

Line 346: Since the treatments UTC and fASD together with MSM were evaluated in two separate areas (UCSC-CfA and PSI), ¿how the authors are sure that when comparing the treatments located in UCSC-CfA with those located in PSI, there are no confounding effects, which can be attributed to differences between the two experimental areas and not to the treatments?

Unfortunately, we cannot be certain. The lack of replication across sites and years is a huge area of concern in this paper. I have gone through and revised several sections in the Abstract, results, and discussion in an effort to reduce over emphasis on across site comparisons of the unreplicated treatments. I have also attempted to revise several sections to include acknowledgement of the lack of replication and more clearly address parts of the analysis where there are high levels of uncertainty in our observations.

Line 347: Did you mean Figure 3 a and 3 b?

Yes, I have corrected this

Line 352: Again,  how the authors are sure that when comparing the treatments located in UCSC-CfA (MSM) with those located in PSI (fASD and FUM), there are no confounding effects, which can be attributed to differences between the two experimental areas and not to the treatments?

fASD and MSM both took place at UCSC-CfA while bASD, UTC, and FUM took place at PSI in both 2019 and 2021. In the discussion section we further discuss the uncertainty of these results as well as observed difference between sites and the challenges we experience with employing each ASD method. A statistical comparison was made because both sites used the same strawberry variety, but I have added context regarding the lack of replication and that impact of site cannot be statistically separated from the impact of treatment. I also clarified that PSI 2021 the strawberry varieties were changed by the grower to a Fusarium wilt resistant variety for both the bASD and UTC.

Line 360: How the authors are sure that when comparing the treatments located in UCSC-CfA with those located in PSI 2019, there are no confounding effects, which can be attributed to differences between the two experimental areas or time periods and not to the treatments?

Please see my above response. I agree completely and have revised several sections throughout the manuscript to better acknowledge these uncertainties and reduce the emphasis on comparing across sites. However, there are other aspects of bASD that we feel reduced its efficacy related to challenges with employing this method in the field. We also discussed how those challenges might be mitigated should this experiment be repeated with appropriate replication of fASD and bASD within sites.

Some additional comments are included in the attached file.

I have read and addressed these comments.

Reviewer 3 Report

The study fills in important gaps related to the volatile organic compounds profile in soils under anaerobic conditions, and the findings will help to advance knowledge of (but not limited to) strawberry disease management. However, my biggest concerns are the comparisons of treatments from different experiments. The study was done in two locations, in different years, and some treatments were not repeated across locations. So, is it really appropriate to compare such treatments? For example, I don't believe fASD (done in location 1 in year 1) should be compared to bASD (from location 2 in year 2). How to disentangle, for example, potential differences intrinsic to treatments from differences because of variability in weather conditions in time (year to year) and space?

I do believe the paper has its value, especially for the reasons pointed out above, but I think authors should be careful with the comparisons and their implications on conclusions.

Please see below a few more suggestions. Authors can also find minor suggestions in the attached pdf file.

Authors should double-check the units throughout the manuscript for consistency because units of time like hours is being expressed in different ways (e.g. "hours", "hrs").

Authors conducted experiments during the 2018-2019, 2019-2020 and 2020-2021 seasons, which are being indicated as UCSC-CfA2018, PSI2019 and PSI2021, respectively. Having this in mind, shouldn't "PSI2021" actually be "PSI2020"?

Abstract

-The treatments are not very clear. The results seem to indicate that treatments are a combination of factors within a) and b).

Abbreviations: The list of abbreviations should be double-checked for consistency. 

- bASD is indicated as "Formed bed ASD" in the list and as "conventional bed" in the abstract.

- UTC is indicated in the abstract and not in the list of abbreviations. Should UTC also be included in the list?

Introduction

-The first and second paragraphs can be merged.

-Authors should add a paragraph to introduce the pathogens and effects on the plants. This is important especially because such impacts are actually being measured in the study. Thus, it is important to build a foundation for a better understanding of readers not so familiar with pathogens.

Methods

- Lines 170 to 173: is this a randomized COMPLETE block design? 12 plots arranged in 4 blocks with 3 treatments per block. Also, is the sample unit size corresponding to 2 beds?

- Why was repeated measures anova used for some variables and linear mixed models for others? Repeated measures, randomized block and split plot designs can actually all be modelled using rm-anova or lmer. However, the more complex the design (which seems to be the case here), the more advantageous it is to use lmer. So, what was the criteria here? Why not just use lmer?

Figures

Figure 2 - why are figure legends in different places in each panel? The majority are at the bottom left... Why not do the same for all of them? Also, for panel f, why use "fumigated" instead of "FUM"?

Figure 3 - why are the y-axis titles in red while all the rest are black? Also, would the use of letters facilitate the comparisons? Furthermore, the figure caption describes differences but it does not mention all of them (e.g., PSI 2021 for yield).

Figure 5 - Same thing with the y-axis titles

Author Response

The study fills in important gaps related to the volatile organic compounds profile in soils under anaerobic conditions, and the findings will help to advance knowledge of (but not limited to) strawberry disease management. However, my biggest concerns are the comparisons of treatments from different experiments. The study was done in two locations, in different years, and some treatments were not repeated across locations. So, is it really appropriate to compare such treatments? For example, I don't believe fASD (done in location 1 in year 1) should be compared to bASD (from location 2 in year 2). How to disentangle, for example, potential differences intrinsic to treatments from differences because of variability in weather conditions in time (year to year) and space?

I have rewritten portions of the abstract and added more detail to the methods and results section regarding our analyses. I agree that we can’t draw strong statistical conclusions about the differences between treatments performed at different sites over different years given the lack of replication, and overall agree with your sentiment. However, this was not clearly reflected in our abstract or throughout the first half of the paper. In the discussion section we address the differences between sites; however, there is too strong of an emphasis on comparing fASD and bASD. I have attempted to revise the discussion to reduce this emphasis.

I do believe the paper has its value, especially for the reasons pointed out above, but I think authors should be careful with the comparisons and their implications on conclusions.

Please see below a few more suggestions. Authors can also find minor suggestions in the attached pdf file.

I have gone through the attached pdf and made any small changes that were noted.

Authors should double-check the units throughout the manuscript for consistency because units of time like hours is being expressed in different ways (e.g. "hours", "hrs").

I have gone through and attempted to correct this.

Authors conducted experiments during the 2018-2019, 2019-2020 and 2020-2021 seasons, which are being indicated as UCSC-CfA2018, PSI2019 and PSI2021, respectively. Having this in mind, shouldn't "PSI2021" actually be "PSI2020"?

Experiments were conducted from 2018-2019, 2019-2020, and 2021-2022. Fall of 2020 was not season in which measurements were conducted due to the COVID-19 pandemic and other logistical barriers. There was typo in the first sentence of the section titled “Formed Bed ASD Field Trials” which erroneously dated the PSI 2021 experiment as 2020-2021.

Abstract

-The treatments are not very clear. The results seem to indicate that treatments are a combination of factors within a) and b).

I have rewritten the abstract to more explicitly describe what treatments are associated with which sites and removed the confusing a and b notation.

Abbreviations: The list of abbreviations should be double-checked for consistency. 

- bASD is indicated as "Formed bed ASD" in the list and as "conventional bed" in the abstract.

- UTC is indicated in the abstract and not in the list of abbreviations. Should UTC also be included in the list?

UTC should be included and was left out by mistake. I fixed the table but another reviewer suggests to remove it and it is not required by Agronomy

Introduction

-The first and second paragraphs can be merged. Done

-Authors should add a paragraph to introduce the pathogens and effects on the plants. This is important especially because such impacts are actually being measured in the study. Thus, it is important to build a foundation for a better understanding of readers not so familiar with pathogens.

I expanded the third paragraph to briefly describing the pathogens and their impacts on strawberries in the central coast.

Methods

- Lines 170 to 173: is this a randomized COMPLETE block design? 12 plots arranged in 4 blocks with 3 treatments per block. Also, is the sample unit size corresponding to 2 beds?

Yes, this was a randomized complete block design within each site as described in the reviewer’s comment. However, the sample size for the VOC monitoring would only apply to one bed not both as the tubing was only installed in the one bed. This bed is also the same bed in which wilt score, mortality, and yield were monitored. I have revised this section of the methods to better communicate this.

- Why was repeated measures anova used for some variables and linear mixed models for others? Repeated measures, randomized block and split plot designs can actually all be modelled using rm-anova or lmer. However, the more complex the design (which seems to be the case here), the more advantageous it is to use lmer. So, what was the criteria here? Why not just use lmer?

I have rewritten the data analysis section and clarified that this section to explain some of the decisions we had to make with the gas data which proved to be difficult to analyze. We did standardize the gas data from the MSM plots to compare to the gas data from the fASD plots but we did not do this for the UTC and bASD. This was because the static chamber method was extreme time prohibitive. We continued to sample the MSM throughout the study but biweekly rather than weekly and both treatments could not always be sampled on the same day due to time constraints. Additionally, we did an across site analysis of gas concentrations comparing bASD and fASD, but with a within treatment variable for Eh Phase (whether the plot was aerobic or anaerobic at time of sampling) and compared the difference in gas concentrations between phases. I have rewritten the section to better describe this analysis which was not well described before, and acknowledged the lack of control for confounding variables as well as other considerations that reduce confidence in across site comparisons.

Regarding RM-ANOVA vs lmer: My understanding is that there some discourse on the validity or accuracy of p-value interpreted from models using the lmer function in r; however, much of the voc data in particular does not meet the assumptions of RM-ANOVA due to missingness or zero-inflation necessitating lmer for any amount of quantitative analysis. I was uncertain how best to approach this and elected to use the RM-ANOVA when the assumptions were met and lmer as the non-parametric alternative when they were not. The vast majority of the VOC data was analyzed using lmer with a handful of GHGs analyzed with RM-ANOVA.

Figures

Figure 2 - why are figure legends in different places in each panel? The majority are at the bottom left... Why not do the same for all of them? Also, for panel f, why use "fumigated" instead of "FUM"?

Fixed, they should all be in the same location

Figure 3 - why are the y-axis titles in red while all the rest are black? Also, would the use of letters facilitate the comparisons? Furthermore, the figure caption describes differences but it does not mention all of them (e.g., PSI 2021 for yield).

Figure 5 - Same thing with the y-axis titles

Made all axes black. The results from PSI 2021 for these data really aren’t comparable. At the very least for UCSC-CfA and PSI 2019 we had the same strawberry varieties to compare wilt, mortality, and yield. However, the grower switched varieties for PSI 2021 to specifically grow resistant varieties in the bASD and UTC due to labor shortages and concerns over poor yield. We described this in the methods but I have rewritten the results and captions to better acknowledge those concerns.

Round 2

Reviewer 2 Report

The authors carefully made the suggested modifications and make clear the limitations that in some cases the absence of repetitions generated, as an element to take into account in the interpretation and recommendation of the results. I consider that with these changes the article is ready to be published. 

Author Response

Thank you so much to the reviewer for your comments. They have been extremely helpful during this process.

Reviewer 3 Report

Authors did a really good job addressing all the comments and concerns. I just have a few minor suggestions below:

Abstract (line numbers with activated track changes)

line 22: GHGS needs to be defined as this is the first mention to it.

line 25: should the unit for mortality (%) be indicated within the parenthesis?

line 30: the unit for yield is being represented by t/ha in the figures. Should the unit in the abstract and throughought the manuscript be chnged to "t/ha" for the sake of consistency? Speaking of which, "t/ac" is used is multiple instances in the manuscript. Should this also be changed to "t/ha"?

line 35: "observations"

Introduction

line 76: I believe "V." should be "Verticillium" because it is the at the beginning of a sentence.

Methods: "in place"?

Author Response

Thank you so much to the reviewer for your comments. They have been extremely helpful during this process.

Abstract (line numbers with activated track changes)

line 22: GHGS needs to be defined as this is the first mention to it.

done.

line 25: should the unit for mortality (%) be indicated within the parenthesis?

Good point, the abstract doesn't indicate any units so I added to the parenthesis as suggested

line 30: the unit for yield is being represented by t/ha in the figures. Should the unit in the abstract and throughought the manuscript be chnged to "t/ha" for the sake of consistency? Speaking of which, "t/ac" is used is multiple instances in the manuscript. Should this also be changed to "t/ha"?

I double checked the yield units and they should all be t/ha now. The application of the Carbon sources was done and reported in tons/acre (t/ac). I have typically seen C source application rate reported in t/ac in the literature and I feel it's a more accessible unit for growers even if it isn't SI. I realize this could be confusing given "tonnes" vs "tons" both being represented as "t" so I have defined the units in the abstract more explicitly.

line 35: "observations"

fixed.

Introduction

line 76: I believe "V." should be "Verticillium" because it is the at the beginning of a sentence.

fixed.

Methods: "in place"?

fixed